# Developmental differences in children's and adults' use of geometric information in map-reading tasks

Yenny Otálora[1]*, Hernando Taborda-Osorio[2]

1 Center for Research in Psychology, Cognition and Culture, Institute of Psychology, Universidad del Valle, Cali, Colombia, 2 Department of Psychology, Pontificia Universidad Javeriana, Bogotá, Colombia

☯ These authors contributed equally to this work.
* yenny.otalora@correounivalle.edu.co

**Data Availability Statement:** All relevant data are within the manuscript and its Supporting Information files.

## Abstract

Using maps effectively requires the ability to scale distances while preserving angle and orientation, the three properties of Euclidean geometry. The aim of the current study was twofold: first, to examine how the ability to represent and use these Euclidean properties changes with development when scaling maps in object-to-object relationships and, second, to explore the effects on the scaling performance of two variables of the array of objects, type of angular configuration and relative vector length. To this end, we tested seventy-five 4-, 6-, and 8-year-old children, as well as twenty-five adults, in a simple completion task with different linear and triangular configurations of objects. This study revealed important developmental changes between 4 and 6 years of age and between 8 years of age and adulthood for both distance and angle representation, while it also showed that the configuration variables affected younger and older children's performances in different ways when scaling distances and preserving angles and orientation. This study was instrumental in showing that, from an early age, children are able to exploit an intrinsic system of reference to scale geometrical configurations of objects.

## Introduction

Understanding maps is one of the most important large-scale spatial abilities humans have to master to find objects and locations in real-world space. Maps are tools that depict spatial symbols that allow humans to represent multiple relationships between objects and locations that could be either directly perceived or unperceived. According to Uttal and Sheehan [1], maps constitute cultural artifacts that provide people with a mediated perspective of large-scale space, which is valued knowledge that is transmitted and accumulated through generations. Therefore, investigating how children develop the ability to represent and use geometric information to read maps effectively is an important goal in the study of cognitive development [2, 3].

Using maps to navigate and find objects in a three-dimensional environment implies three different cognitive abilities. First, using maps requires establishing the *symbolic* correspondence between the map and the real space. Thus, a map reader should understand the one-to-one relationship between the symbols depicted in the map and the objects in the three-

**Funding:** This is a Research Article product of the research projects "Innovation in educational intervention for the development of mathematical thinking in early childhood" code 1203-1003-72878, funded by Mineducación, Minciencias and Dividendo por Colombia, awarded to YO and HT and "Prediction in young children and social behavior", code 00007819, funded by Pontificia Universidad Javeriana, Bogotá-Colombia, awarded to HT.

**Competing interests:** The authors have declared that no competing interests exist.

dimensional spatial layout [4, 5]. Second, using maps requires understanding the *geometric* correspondence between the map and the real space. Therefore, a proficient use of a map either to place or to find objects entails understanding how the objects' geometric position in the map is preserved in the three-dimensional spatial layout [6]. Third, since the map and the depicted three-dimensional space are usually different sizes, the map reader should scale the distance information represented in the map [7]. Hence, if in the map of a mall, the relevant location is 4 inches from the main entrance, and the scale of the map is 1:50, then we have to walk 200 inches from the entrance to reach the target. Spatial scaling requires from the map reader to understand both the symbolic and the geometric correspondence, but it adds the ability to mentally transform distances between spaces with different sizes [8].

Previous research on the development of map reading abilities has shown that as early as 2 ½ years of age, children demonstrate a basic understanding of both symbolic and geometric correspondence [9, 10]. For example, 30-month-olds, but not younger children, are capable of using pictures (a photograph and a line drawing) to find a hidden object in a large real room based on symbolic correspondences (e.g., the hidden Snoopy toy is behind the chair; 9). Similarly, 30-month-old children are able to use a purely geometric map to place a toy in triangular and linear object arrays [10]. Children are especially proficient at distinguishing locations based on some categorical spatial information, such as the apex of the triangle and the "in between objects" relationship. Older children have been shown to represent different types of basic geometric information in map-reading tasks. Namely, children are able to represent and use distance, angle and sense information in geometric maps. Thus, 4-year-olds are able to use distance information in an array of unconnected dots depicting linear and triangular configurations to place objects in the correct positions [11]. However, children this age have been shown to have a better performance when researchers used a pattern of connected dots with a distinctive triangular shape than when using a pattern of unconnected dots [12, 13].

The use of angular information in maps in young children has been more controversial as angular information correlates with relative distance information in triangular configurations. When using an isosceles triangle configuration 4-year-olds show low sensitivity to either angular or distance information [11]. In trying to disentangle both sources of information more recent research has shown that preschoolers are also able to use angle information alone to locate objects in the correct position [14, 15]. However, the research also reveals a developmental trend with 6-year-olds been significantly more proficient at using angular information than 4-year-olds (63% compared to a 50% chance level; 12). Although preschoolers show sensitivity to distance and angular information, they also show clear limitations when distinguishing the correct location based on orientation information alone. For instance, in an isosceles triangle, children place objects at chance between the two locations in the left and right sides of the apex [11].

Another line of research [16, 17] has demonstrated that the ability to extract geometric information from maps is supported by two evolutionary systems of non-symbolic spatial representations: a system of layout geometry and a system of object geometry [18]. The first system -layout geometry- supports spatial navigation and uses orientation and distance information to represent locations in the environment; the second system -object geometry- supports object recognition and uses distance and angle information to represent shapes. The studies show that children use both systems to interpret the geometric information on maps, but along development children become increasingly skillful at using distance and angle information in a more integrated fashion [17].

In spite of the early achievements in extracting symbolic and geometric information from simple maps and pictures, scaling spatial information seems to emerge later and through a protracted developmental process. Thus, when shown a simple one-dimensional map (i.e., a

narrow rectangular enclosed space), 60% of 3-year-olds and all 4-year-olds succeeded in placing an object in the correct position of a larger sandbox [19]. However, in another study, when the children were asked to utilize a two-dimensional map (i.e., a wide rectangle), the pattern of successful achievement was delayed by one year, and thus 60% of 4-year-olds and 90% of 5-year-olds succeeded at the placement task [20]. This overall developmental trajectory was further supported in another study [7]. Rather than asking participants to place an object into a sandbox, researchers presented two spatial layouts of different sizes on a sheet of paper and asked participants to use the smallest layout (i.e., the map) to place an object on the larger two-dimensional space in the correct position. The findings of this study showed a significant improvement in scaling abilities between 3 and 5 years of age and nonsignificant differences beyond that age point, including adult participants. However, contrasting results were reported in a more recent study [8]. By using a computer-assisted discrimination task where participants were asked to distinguish the correct scaled array of a referent map from several target configurations, it was found that the children's ability to scale continued to develop until 8 years of age and became increasingly precise.

From the previous studies, it is apparent that reading maps to scale distances is an early emerging ability that requires participants to encode distance relative to an extended surface either in a one- or two-dimensional space. However, the use of this type of experimental setups makes unclear how participants can exploit all available geometric information in the array of objects to solve the task. For instance, in real-life situations, as when reading the campus map, people use distance, angle and orientation information among points and landmarks in the map to guide their search while scaling the correct distance to walk. To our knowledge, Uttal [21] carried out the only study partially addressing this issue about the children's use of geometric information in the context of a scaling task. In that study, the performance of 4- to 5-year-olds, 6- to 7-year-olds and adults were compared in a reconstruction task of a relatively complex configuration of 6 objects. Each participant was asked to memorize the configuration and then try to reconstruct it in a different larger room. The results of this study revealed, first, that even preschoolers were able to preserve the relative relations among the objects, meaning that they placed them in a similar ordinal correspondence. Second, preschoolers were overall less accurate when scaling absolute distances and when preserving angular relations among the objects compared to elementary school children. However, this difference was mainly explained by the preschoolers' inferior performance when scaling the large configuration of objects. Third, preschoolers also exhibited inferior performance compared to other children when scaling an asymmetrical configuration. These findings show that preschoolers' behaviors vary to a large extent as a function of variables of the configuration to be scaled. In particular, size and symmetry seem to be important factors moderating the accuracy of the preschoolers' performance.

Uttal's [21] study shows some developmental differences when scaling a configuration of objects. However, a shortcoming in this study appears when examining the sources of the young children's difficulties in scaling. An object's relative position in a configuration depends on distance, angular relationship and orientation, but in Uttal's reconstruction task, it is difficult to say how all these dimensions are differentially weighted by each participant. For instance, preschoolers may find it more difficult to preserve angle information than scaling distances, which may prevent them from accurately reconstructing the array of objects.

## Current study

The current research aims to gain further insight into the developmental changes of how children and adults use geometric information within the context of a scaling task. To this aim,

participants in our study solved a simple completion task through different linear and triangular configurations of objects. In each configuration, participants observed a purely geometric map with only a three-point array, which represented the objects in a larger three-dimensional space. One of the points on the map signaled the correct relative location of a target object in the three-dimensional space where only the other two nontarget objects were already in place. Each participant was asked to use the map to place the target object in the correct location in the referent space. To solve the task, participants should determine the target object's position relative to the two reference points and complete the object's configuration. The use of purely geometric maps devoid of any landmark allows us to examine how the participants extract and represent the three Euclidian properties from the configuration of objects: distance, angle and orientation. Moreover, this type of completion task allows us to investigate the developmental trajectory for each of these three Euclidian dimensions that participants have to represent when solving the task. The participant's performance in the task may be accurate regarding one of these dimensions but not necessarily in all of them.

In addition to tracing the developmental trajectory of the participants' use of geometric information in a map-reading task, the second main objective of the current study was to examine how differences in both the type of angular configuration of objects and the relative vector length affect the participants' performance in the scaling task. The first variable, *configuration type*, was operationalized as the angular difference between two of the vectors of the configuration formed by the three objects (180˚, 135˚, 90˚ and 45˚), one of which was previously defined as a reference vector and the other as the vector the participants have to reconstruct. Because the configuration of three objects does not have "sides", it is more precise to label this quantity as a "vector", having magnitude and direction, between two objects of the configuration in the three dimensional space. By manipulating this variable, we wanted to see, first, whether the participants' ability to preserve angle information in the completion task varied when opened (135˚) and closed (45˚) vectors were reconstructed, and second, whether the participants' ability to scale distances varies when the reconstructed vector of the configuration is either on the same (180˚) or on a different (90˚) axis from the other vector of the configuration.

The second variable, *relative vector length*, was operationalized as the proportion between the reconstructed vector (e.g., 115.2 cm. for the longest vector) and the sum total of both the reconstructed and reference vectors, which was kept constant across maps (144 cm.). Four lengths were utilized: Length 1 (115.2/144 = 0.8), Length 2 (93.6/144 = 0.65), Length 3 (50.4/144 = 0.35) and Length 4 (28.8/144 = 0.2). By manipulating this variable, we wanted to examine the possible differences in accuracy when the reconstructed vector is shorter than the reference vector (Lengths 3 and 4) versus when the reconstructed vector is larger (Lengths 1 and 2).

We chose these configuration variables because they are basic spatial dimensions in which one vector may differ from another vector in a completion task. Distances in a configuration of objects could be longer or shorter compared to a distance of reference and they can have different angular relationships. None of these variables have been explored before in the context of a scaling task. Previous research has demonstrated that young children are able to distinguish different angular configurations [14], but it is unknown whether in a scaling task some angular configurations are easier to preserve than others. Moreover, the typical scaling task occurs in enclosed spaces [19, 20] where the reconstructed vector is always inside of the reference rectangular frame. However, by using a scaling task in an object-to-object configuration we can explore the participant's performance in the alternative scenario where the reconstructed vector is larger than the reference vector. To this aim, we examined the effect of both configuration variables: configuration type and relative vector length. By addressing the effect of these variables, we can obtain a more detailed understanding of how children from different

age groups and adults use the geometric information when scaling distances. Therefore, we are interested not only in determining at what point in development children succeed in exploiting different types of geometric information in an array of objects but also in determining under what conditions they do better or worse.

## Methods

### Participants

Seventy-five children from three age groups participated in the study: 4-year-olds (N = 23, Mage = 52 moths, range = 49–59 months; 13 girls), 6-year-olds (N = 26, Mage = 77 months, range = 72–81 months; 12 girls) and 8-year-olds (N = 26, Mage = 101 months, range = 96–104 months; 12 girls). Additionally, 25 adults participated in this study (Mage = 20 years, range = 19.6–20.6 years; 14 women). This group consisted of psychology students at the undergraduate level. Both children and adults came from racially mixed, urban, middle-class families.

Ethical approval for the study was obtained from the Institutional Review Committee for Human Research Ethics and Animal Experimentation of the Universidad del Valle. For the children, written informed consent was obtained from their parents and written informed assent was obtained from the children before data collection. Children were thanked for their participation with a small souvenir. Moreover, written informed consent was obtained from the psychology students before data collection. They were asked to participate in the study for course credit.

### Materials

The completion task consisted of 12 purely geometric maps, each map representing a specific linear or triangular configuration of three objects. Each configuration was represented on the map with three blue ink dots 6 mm. in diameter. The dot of the configuration located at the coordinates 0,0 (in an x-and-y system) was called *point 0*, and a black ink X of 6 mm. x 6 mm. was placed on one of the other two dots, representing the *target location* (Fig 1). The vector formed between point 0 and the target location was called the *reconstructed vector*, because it is the quantity the participants have to reconstruct in the three-dimensional space during the task, and the vector between point 0 and the other reference point was called the *reference vector*, because it is the quantity the participants observe in the three-dimensional space and have as the only reference to locate the target object during the task. The sum total of the lengths of these two vectors was always 12 cm. in the maps. The measures on the map were as follow: Length 1 (reconstructed vector = 9.6 cm.), Length 2 (reconstructed vector = 7.8 cm.), Length 3 (reconstructed vector = 4.2 cm.) and Length 4 (reconstructed vector = 2.4 cm.). Each map was printed on a bond-based paper measuring 21.6 cm. high x 28 cm. wide. The reference vector in the open space was built with two wooden toys (sheep 7 cm. high) positioned on the ground. Participants should place a third wooden toy (a lion 7 cm. high) in the correct target location as represented in the map by the X in order to complete the three-object configuration. The scaling factor between the reference vector in the map and the reference vector in the open space was 1:12.

### Design

The design of the maps varied according to two intrasubject variables: (a) configuration type and (b) relative vector length. Four types of configurations of objects were utilized based on the angular grade between the reconstructed vector and the reference vector: linear (180˚; 4

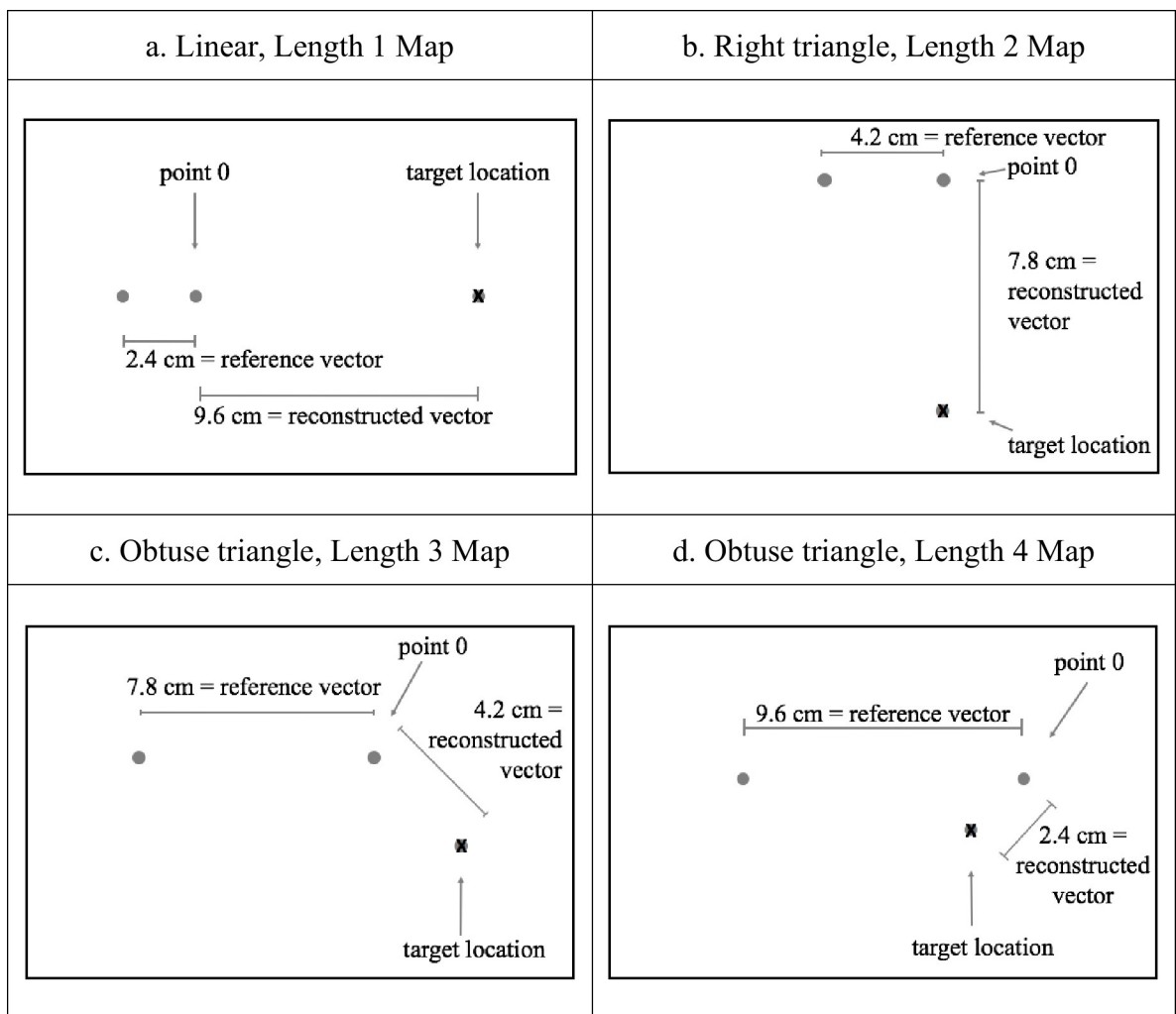

**Fig 1. Sample maps.** Four types of angular configurations and four vector lengths.

maps), right triangle (90˚; 4 maps), acute triangle (45˚; 2 maps) and obtuse triangle (135˚; 2 maps) (Fig 1). For the linear and right triangle configurations, 4 relative lengths were utilized based on the proportion between the reconstructed vector and the sum of both the reconstructed and reference vectors, as explained before: Length 1 (0.8), Length 2 (0.65), Length 3 (0.35), and Length 4 (0.2). For the Acute and Obtuse triangular configurations, only proportions of 0.35 and 0.2 were utilized. We used only these two proportions because Length 1 and 2 for the acute configuration would not be easily interpreted as acute triangles. Two more maps showing configurations of three objects were created to familiarize children with the task; the first was linear and the second was triangular. Both maps had a proportion of 0.82. (reference vector = 3.0; reconstructed vector = 9.0).

## Procedure

The children were tested individually in a large open space located in their schools. There were no visible marks on the ground and no landmarks close to the test space. Each map was presented to the participants on a student table. Each participant stood in front of the table and

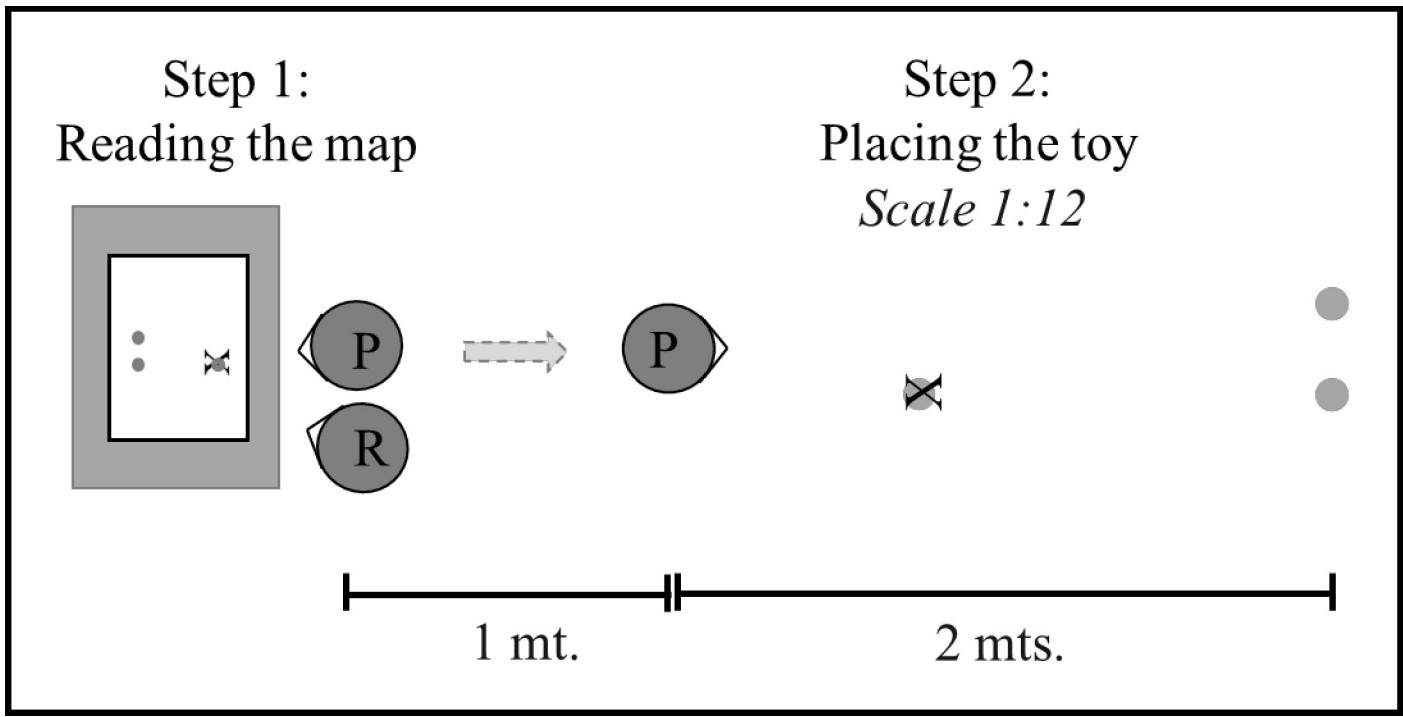

**Fig 2. Experimental set up.** Placement of the participants and the materials for the task.

was able to observe the map from above. The researcher was positioned directly on the left side of the participant in front of the table and was able to observe the map from above and point out the target location on the map. The three-dimensional layout was located behind both of them 3 mt. apart, where only the two sheep were already placed (Fig 2). Each participant was asked to look at the map, and then go to the three-dimensional space and locate the lion in the place represented by an X on the map. The map and the three-dimensional layout were never visible simultaneously for the participant. After looking at the map, the participant had to rotate 180˚ and walk to the place where he/she decided to locate the lion. The orientation of the configuration of objects in the three-dimensional layout relative to the participant had the same orientation than the map relative to the participant. The specific procedure included two phases: the practice phase and the test phase.

**Practice phase.** In the beginning of each practice trial, the researcher first told the child the following story: "Let's go to play with the sheep and the lion. Look (pointing to the map on the table), this picture is the drawing of the space you have behind you. The dots show you the places where the sheep and the lion should be located. The two sheep are already located in the corresponding place; look at the two sheep (holding the child's shoulders and turning him/her around 180˚ in clockwise direction, showing him/her where the sheep were positioned and again turning the child around in clockwise direction to look at the map). However, the lion does not know where it should be located (showing the lion on her hand). This lion should be located here where the X is; could you show me in the picture where the X is? (waiting for the child to point to the X on the map). Well, now take the lion (handing the lion to the child). You should remember the lion's position and go and put the lion in the place where it should be". The researcher gave the child 5 seconds to observe the map. Then, she took away the map and told the child "You can go now". The researcher held the child's shoulders and turned him/her around 180˚ in clockwise direction. Then she walked 1mt straightforward with the

child and stood in front of the configuration of sheep, then she waited for the child to locate the lion. The child was not allowed to walk around the table or take any different direction to that instructed by the researcher. Feedback was given for the two practice trials. If the child's response was close to the target location, he/she was told "you did well". If the child's response was far from the target location, he/she was asked to look at the map again and repeat the trial. At the end of each practice trial, the correct response was shown to the child, and the researcher pointed to the correspondence between the correct location in the three-dimensional space and the target location on the map, emphasizing that the dots on the picture indicated where the animals were located.

**Test phase.** Test trials were similar to practice trials except that a summarized prompt was given to the children, the researcher showed the map to the children just once at the end of the prompt and they received no feedback on their performance during the trials. In the beginning of each test trial, the child was told "Look now at this picture. The picture shows you where the sheep and the lion should be located in the space behind you. Now take the lion (handing the lion to the child). You should remember the lion's position; go and put the lion on the place where it should be". The researcher gave the child 5 seconds to observe the map and took it away. The researcher held the child's shoulders and turned him/her around 180° in clockwise direction. Then, the child walked alone 1mt straightforward as instructed, stood in front of the configuration of sheep and located the lion on the ground. The researcher waited close to the student table for the child to locate the lion. Then the researcher put the next map on the table for a new trial. During the test phase all 12 maps were presented in random order for each participant. The experiment lasted approximately 15 min. The adults were tested individually in a large open space at their university campus, without visible marks or landmarks close to the test space. The procedure was the same as described for children during both the practice phase and the trial phase.

## Data collection

To investigate the participants' accuracy in representing distance during the completion task, the deviations of their responses from the target locations were analyzed as absolute distances in centimeters. To examine participants' accuracy in representing angle during the completion task, the deviations of their responses from the target locations were analyzed in degrees. Finally, to investigate participants' accuracy in representing orientation during the completion task, their performance was classified according to either left-and-right reversals or up-and-down reversals.

Measurements were performed during the experimental tests. After each response, the researcher marked the position of the lion by placing a sticker on the ground. The x-and-y coordinates of this position were recorded, considering the point 0 connecting the reference vector and the reconstructed vector. To record the position of the lion and the point 0 (determining the child's reconstructed vector), the researcher first located a hard metal meter passing by the reference points and a letter-sized white bond paper adjacent to it. Then, she took two pictures of each configuration of objects from a height of 1.80 m. To determine and codify the exact distance and angle deviations as well as orientation reversals, these pictures were analyzed with the high-quality measurement software iPhotoMeasure [22].

## Data analysis

Each variable was analyzed for outliers, and values that were more than three standard deviations above or below the mean were excluded (6 data points in total or 0.5%). The results were analyzed in three sections: distance, angle and orientation. For distance, only the linear and

right configurations were included in the omnibus analysis of variance (ANOVA). This is because only these configurations have all four Lengths to compare the effect of short versus long reference vectors. Due to some of the distance variables having right-skewed distributions, analyses were conducted with bootstrapping [cf. 23]. For angle, only the acute and obtuse configurations were included in the omnibus ANOVA. This is because these configurations have only two lengths. Based on Q-Q plots, no serious deviations from normality were observed. For orientation, all four configurations were analyzed with nonparametric statistics.

## Results

### Analyses of distance

Preliminary analyses found no effect of Sex; therefore, this variable was collapsed in further analyses. A 4 (Age: 4, 6, 8, and Adults) X 2 (Configuration Type: Linear and Right Triangle) X 4 (Length: 1, 2, 3 and 4) mixed- design analysis of variance (ANOVA) revealed a significant main effect of Age, $F(3, 93) = 18.2$, p < .01, $\eta^2 p = .37$, which resulted from less error in the completion tasks for the Adults (M = 13.1, SD = 5.4) than in any other age group, namely, 4-year-olds (M = 27.4, SD = 7.72), 6-year-olds (M = 24.3, SD = 7.95), or 8-year-olds (M = 24, SD = 7.6). Accordingly, statistically significant differences were found only between the Adults and 8-year-old children, $t(43.3) = 5.8$, p < .01, d = 1.6. The main effect of Length was also significant, $F(2.6, 242.5) = 12.9$, p < .01, $\eta^2 p = .12$, with participants being less accurate with the longest vectors (M = 27.9, SD = 17.3, for Length 1; M = 22.2, SD = 13.6, for Length 2; M = 21.3, SD = 13.5, for Length 3, and M = 17.4, SD = 12.5, for Length 4). There was not a reliable effect of Configuration Type, $F(1, 93) = 3.48$, p = .065, $\eta^2 p = .03$, (M = 23.41, SD = 10.83, for the Lineal Configuration, and M = 21, SD = 10.07, for the Right Configuration).

There was no significant interaction between Age and Configuration Type, $F(3, 96) = 1.66$, p > .1, $\eta^2 p = .07$. However, the ANOVA revealed a significant interaction between Age and Length (Fig 3), $F(9, 279) = 5.28$, p < .01, $\eta^2 p = .15$, that was followed with post-hoc analysis. For the longest Lengths (1 and 2), there were significant differences in errors between the 4-year-olds and all other age groups, (all ps < .01), and between 8-year-old children and Adults, (all ps < .05, except for Length 1 which was only marginally significant, p = .054). Additionally, the Adults significantly differed from 8-year-old children in Lengths 3 and 4 (all ps < .05).

A within-subject ANOVA across the 4 lengths for each age group revealed significant differences only for 4- and 6-year-old children, $F(3, 60) = 17$, p < .01, $\eta^2 p = .46$, and F(3, 53.7) = 3.16, p = .047, $\eta^2 p = .11$, respectively. Post hoc analysis showed significant differences between the shortest and the largest vector only for 4-year-olds, (p < .01). The ANOVA also yielded a significant interaction between Configuration Type and Length, $F(2.7, 256.1) = 3.02$, p = .03. Post hoc analysis revealed significant differences between Linear and Right configurations only for Length 2, with more errors in the linear configuration (M = 25.7, SD = 18.97, and M = 18.75, SD = 16.8, respectively), p < .01.

Finally, we determined the proportion of children who preserved the ordinal relations across the 4 lengths for each of the two types of configurations, namely, Linear and Right (i.e., children who made no more than 1 ordinal error). For the Linear configurations, 92% of Adults, 88.5% of 8-year-olds, 92.3% of 6-year-olds, and 74% of 4-year-olds preserved the ordinal relations. Although a considerable proportion of 4-year-old children seemed to fail to preserve the ordinal relations in the Linear configurations, a Fisher's exact test did not detect significant differences among the age groups (all ps. > .1). For the Right configurations, 100% of Adults, 88.5% of 8-year-olds, 92.3% of 6-year-olds, and 61% of 4-year-olds preserved the ordinal relations. A Fisher's exact test revealed significant differences between 4- and 6-year-

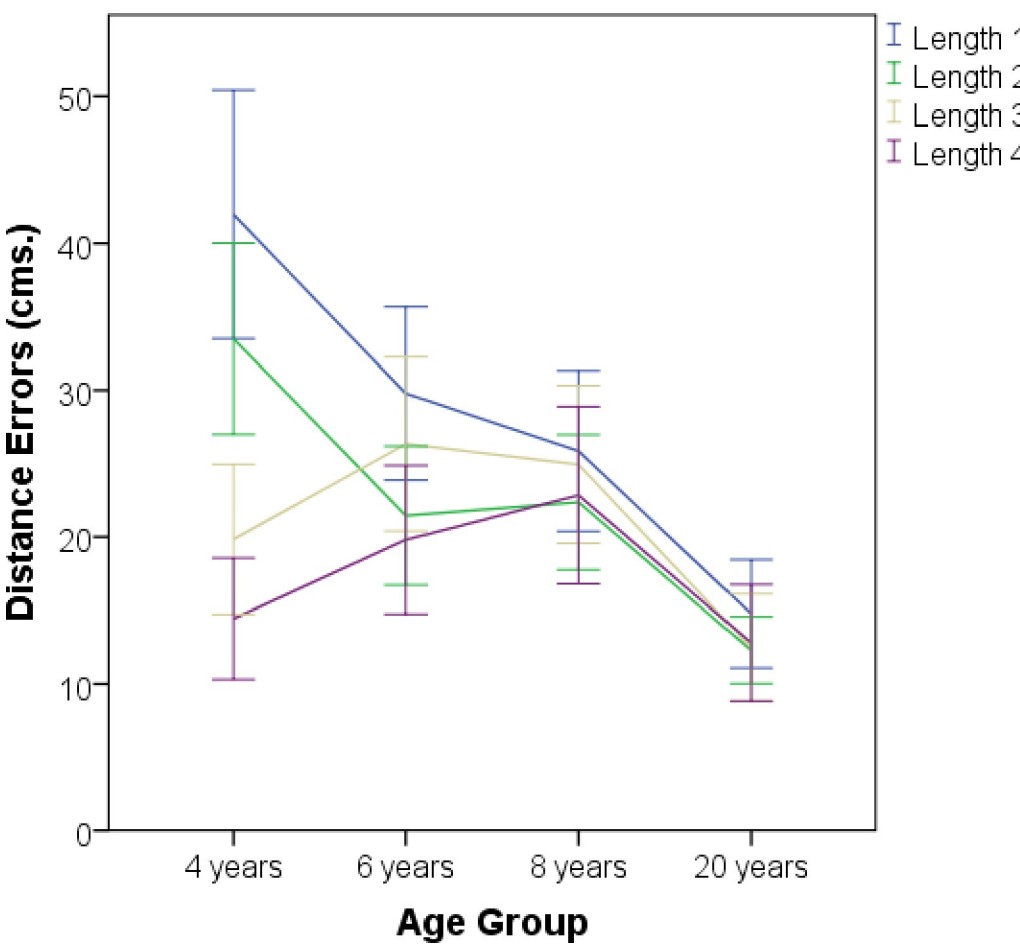

**Fig 3. Mean error for distance.** Error bars represent standard error of the mean.

old children (p = .01) but not among the other age groups (all ps. > .1). These findings indicate that 4-year-olds exhibited inferior performance on the scaling tasks compared to the other age groups. However, the findings also show that a majority of children in the youngest age group still succeeded in preserving the ordinal relations.

Overall, these results show similar performance on the scaling task across all three age groups of children for the two shortest lengths in the configuration. In contrast, 4-year-old children performed worse with the two longest configurations compared to all other age groups. Additionally, the Adults performed much more accurately than children across all eight configurations. Surprisingly, at the shortest Length, 4, the performance of 4-year-old children was not significantly different from the performance of the Adults, $t(46) = .46$, p = .64, d = .04.

## Analyses of angle

Preliminary analyses found no effect of Sex. A 4 (Age: 4, 6, 8, and Adults) X 2 (Configuration Type: Acute and Obtuse) X 2 (Length: 3 and 4) mixed-design ANOVA revealed a significant main effect of Length, $F(1, 95) = 8.4$, p < .01, $\eta^2 p = .08$, which resulted from greater error at the shortest Length, 4 (M = 16.05, SD = 7.9 for Length 3, and M = 18.1, SD = 9.8 for Length 4). The main effect of Age was also significant, $F(3, 95) = 20.2$, p < .01, $\eta^2 p = .39$. Statistically

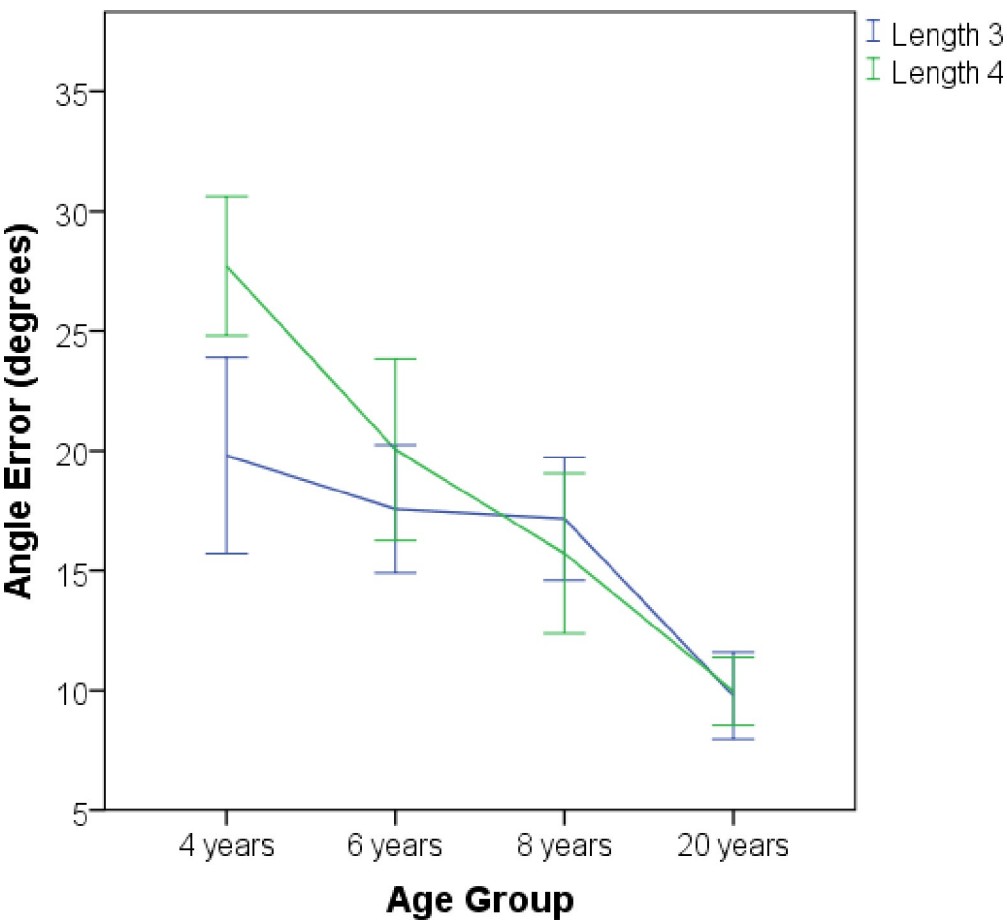

**Fig 4. Mean error for angle.** Error bars represent the standard error of the mean.

significant differences were found only between 4- and 6-year-olds, $t(47) = 2.46$, p = .05, d = .7, with 6-year-olds exhibiting better performance (M = 18.8, SD = 6.8, for 6-year-olds, and M = 23.75, SD = 7.24, for 4-year-olds), and between 8-year-olds and Adults, $t(37.5) = 4.48$, p < .01, d = 1, with Adults exhibiting better performance (M = 9.9, SD = 3.37, for Adults, and M = 16.4, SD = 6.7, for 8-year-olds). However, the ANOVA did not reveal a reliable effect of Configuration Type, $F(1, 96) = 3.1$, p = .08, $\eta^2 p = .04$.

The ANOVA yielded a significant interaction between Age and Length (Fig 4), $F(3, 95) = 6.5$, p < .01, which was followed with post hoc analysis. For Length 3, there were only significant differences between Adults (M = 9.8, SD = 4.3) and all other age groups (all ps < .01). For Length 4, there was a significant difference between 4- and 6-year-olds (M = 27.7, SD = 6.9, and M = 20.05, SD = 9.6, respectively), p < .01, and between 8-year-olds and Adults (M = 15.7, SD = 8.5, and M = 9.8, SD = 3.5, respectively), p = .04. Post hoc analysis between both lengths across all 4 age groups revealed only a significant difference for 4-year-olds, (p < .01). These results show that 4-year-old children differed in the accuracy of their angle representations from the other two groups of children only when they tried to scale a short vector.

Furthermore, the ANOVA also yielded a significant interaction between Age and Configuration Type, $F(3, 95) = 2.87$, p = .04, $\eta^2 p = .08$, which was followed with post hoc analysis. For the Acute Triangle, there were only significant differences between Adults (M = 10.1, SD = 4.7) and all other age groups (all ps < .05). For the Obtuse Triangle, there were significant

differences between 4- and 6-year-olds (M = 27.3, SD = 7.8; M = 19.1, SD = 10.4, respectively), p < .01, between 4- and 8-year-olds (M = 16.4, SD = 8.9), p < .01, and between 8-year-olds and Adults (M = 9.8, SD = 4.4), p = .03. Post hoc analysis between both types of configurations across all 4 age groups showed only a significant difference for 4-year-old children, (p < .01). These results reveal that 4-year-old children performed poorly on the Obtuse Triangle configuration compared to their performance with the Acute Triangle configuration. Children of this age also exhibited poor performance in the Obtuse configuration compared to the performance of the other two groups of children.

Overall, the prior findings reveal that Configuration Type and Vector Length just affected the 4-year-old children's scaling ability, as their performance differed when they tried to scale Acute versus Obtuse configurations and Short versus Long vectors. However, these analyses do not reveal to what extent participants failed or succeeded in representing angle information in the scaling task. To gain a better understanding of this issue, participants were classified as having failed or succeeded in representing angle using an arbitrary cutoff. A reasonable way to classify participant performance representing angle information on the scaling task is by evaluating whether the scaled vector is closer to one of the configuration's intrinsic frames of references (vertical or horizontal) or to the ideal vector (45 g). The rationale is that participants who scaled vectors close to the intrinsic reference frame have a weak representation of angle information. In contrast, participants who tried to scale the vector close to the ideal vector are said to succeed in preserving angle information. A participant was classified as succeeding in the task if the scaled vector was in a range between 25 degrees higher and 25 degrees lower from the ideal vector. Fig 5 presents the results of this classification.

The results of the proportion of successes and failures show that the majority of Adults and 8-year-olds succeeded in preserving angle information across all 4 configurations, namely, Acute Length 3, Acute Length 4, Obtuse Length 3 and Obtuse Length 4 (Mean success = 98% for Adults and 84% for 8-year-olds). A Fisher's exact test confirms the lack of significant differences between these two age groups in all configurations (all ps > .1) except for Obtuse Length 4 (p = .023, two-tailed test). A larger proportion of 6-year-olds failed to preserve angle information across all 4 configurations (Mean success = 70%), but the Fisher's exact test shows no significant differences between 6- and 8-year-old children (all ps > .1). Finally, most of the 4-year-old children failed to preserve angle information in all configurations (Mean success = 52%), except for Acute Length 3 (74%). Four-year-olds significantly differed from 6-year-olds only in the Obtuse Length 3 configuration ($\chi^2$ (1, N = 49) = 4.4, p = .035). However, 4-year-olds significantly differ from 8-year-olds in the Obtuse Length 4 ($\chi^2$ (1, N = 49) = 8.8, p < .01), Obtuse Length 3 ($\chi^2$ (1, N = 49) = 11.8, p < .01), and Acute Length 4 configuration ($\chi^2$ (1, N = 49) = 9.5, p < .01). This analysis reveals a developmental trend with improving performance preserving angle information between 4 and 8 years of age. In agreement with the previous omnibus ANOVA, these results also show that 4-year-old children found it quite difficult to preserve angle information, in particular in the Obtuse configuration, with a low rate of success. Overall, the Acute configuration was easier for children than the Obtuse configuration, with a higher rate of success, even for 4-year-old children.

## Analyses of orientation

The participants' performance in the Linear, Right, Obtuse and Acute Configurations regarding lateral direction were analyzed by comparing correct (left) and incorrect (right) lateral responses (Table 1). The results of each of the 12 configurations was compared to chance through a two-tailed Binomial test. These analyses reveal that, as expected, only the Adults performed significantly different from chance across all Configuration Types (all ps. < .01). In

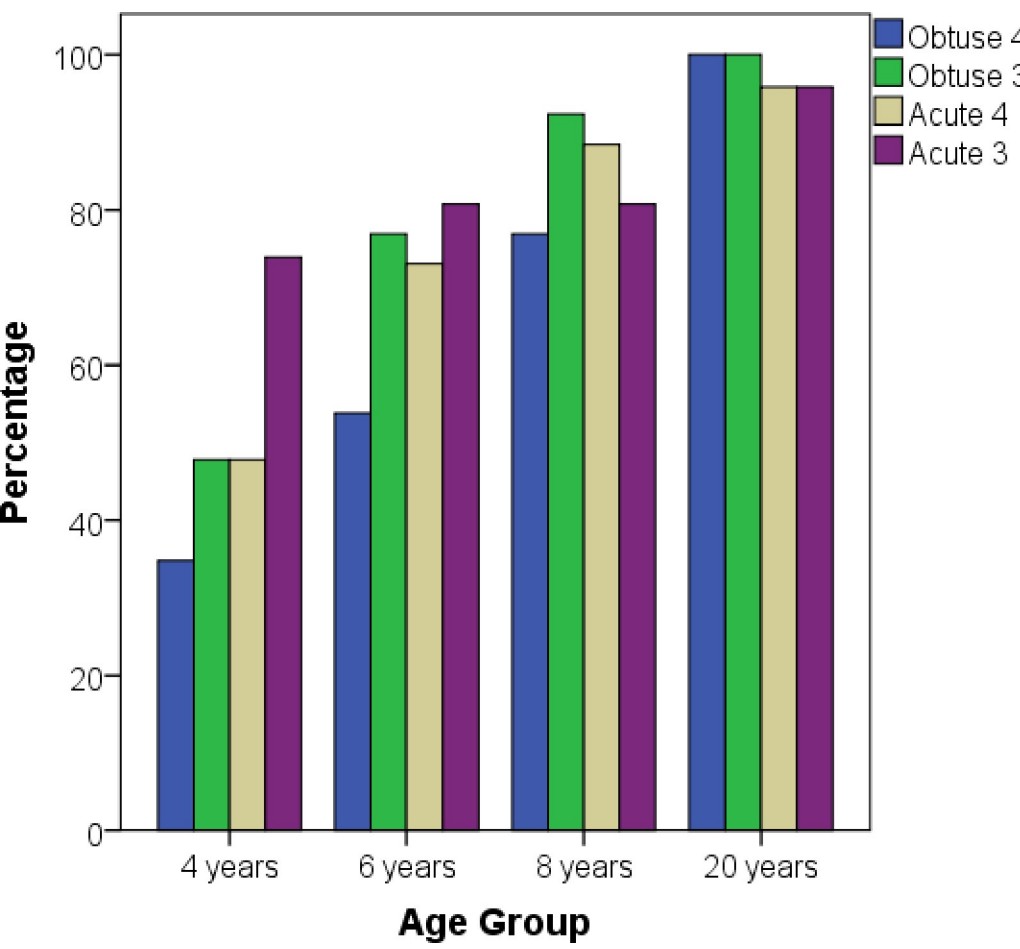

**Fig 5. Percentage of success in preserving angle.**

contrast, 4- and 6-year-olds failed to reach a significant difference from chance (all ps. > .1), suggesting that young children put the toy randomly on the left or right side of the array in all 12 configurations. The results for the 8-year-old children are mixed. These results did not differ from chance across all configurations except in the 3 longest linear arrays (all ps. < .05). In contrast to the lateral direction, the results of the vertical direction in both the Acute and Obtuse configurations show good performance across all 4 age groups. The participants reached a significant difference from chance in all four Configuration Types (all ps. < .05), except for 4-year-olds in the longest linear array (p = .09). These results show that the vertical up-down orientation is easy to represent and use in a scaling task early in development, whereas the lateral right-left orientation is difficult to represent for all the children.

**Table 1. Percentage of correct orientation performance.**

| Age Group | L1 | L2 | L3 | L4 | R1 | R2 | R3 | R4 | O1 | O1V | O2 | O2V | A1 | A1V | A2 | A2V |
|---|---|---|---|---|---|---|---|---|---|---|---|---|---|---|---|---|
| 4 | 65.2 | 52.2 | 61 | 47.8 | 63.3 | 61 | 43.5 | 43.5 | 52.2 | 69.5 | 52.2 | 82.6 | 47.8 | 73.9 | 30.4 | 73.9 |
| 6 | 42.3 | 46.2 | 42.3 | 53.8 | 57.7 | 46.2 | 46.2 | 42.3 | 38.4 | 76.9 | 61.5 | 84.6 | 46.2 | 76.9 | 38.4 | 76.9 |
| 8 | 76.9 | 76.9 | 73.1 | 46.1 | 68 | 50 | 61.5 | 53.8 | 61.5 | 88.5 | 57.7 | 96.2 | 50 | 73.1 | 34.6 | 92.3 |
| 20 | 96.2 | 96.2 | 92.3 | 92.3 | 80.8 | 84.6 | 88.5 | 88.5 | 96.2 | 88.5 | 96.2 | 88.5 | 88.5 | 96.2 | 88.5 | 96.2 |

L = linear configuration; R = right configuration; O = obtuse configuration; A = acute configuration; V = vertical orientation.

## Discussion

The main aim of this study was to investigate the developmental changes in the participants' use of geometric information to successfully scale distances in a map-reading task. The results show evidence for similar developmental trajectories for distance and angle, while orientation shows a relatively late development. The results also reveal differences in the young children's performance across both configuration variables: Configuration type and Relative vector length, suggesting that the children's ability to use the geometric information in the current task depends on each map structure. Below, we discuss the results for each geometric dimension.

In looking only at scaling precision for distance, changes occur mainly in one developmental period, between 8 and 20 years of age. Across all tasks, adults were much more accurate at scaling distances compared to 8-year-old children, meaning that, contrary to all previous findings, the ability to scale distances develops substantially beyond 8 years of age, the point at which children reached a plateau in performance in previous studies [8; 24, 25]. Because we did not test other age groups between 8 and 20-year-olds, we are uncertain as to when in development this change occurs. Although these results show a difference between children and adults in scaling precision this change may occur before adulthood. Future studies should better trace this developmental progression. This result also suggests that scaling distances in object-to-object scenarios is more challenging and complex than scaling in enclosed spaces and, thus, requires more time to develop. Part of this complexity may stem from the fact that scaling distances in object-to-object scenarios could demand a more precise metric representation of the objects' position than scaling in enclosed spaces. Studies on spatial memory [26] have shown than in enclosed spaces an object's position is associated to a subregion in the figure (e.g. in the first quarter of the rectangle) with boundaries that cannot be trespassing. This type of encoding could be more difficult for children and adults to represent in object-to-object scenarios as objects are placed in an open space without visible boundaries. Additionally, previous studies on map-reading tasks show that participants tend to use a mental transformation strategy when scaling distances in enclosed spaces [25], by which distances are mentally expanded or contracted from one space to a different larger or smaller space. The lack of visible boundaries in the current configuration of objects may render the use of this strategy more difficult for all participants.

The results also reveal an interaction between age and vector length. The analyses of distance show that the relative vector length had an effect only for 4- and 6-year-old children. Therefore, 8-year-olds and adults placed the target object with the same accuracy regardless of the vector's length. Surprisingly, the performance of the 4-year-old children did not differ statistically from the adults' performance for the shortest linear and right configurations, whereas adults differed statistically from the other two age groups of children, revealing a classic inverted U-shaped developmental trend. One possibility to interpret these findings is that 4-year-olds may be utilizing relative distances in the two shortest configurations compared to all other participants. Namely, for Length 3 the children might try to place the object to half the distance with respect to the reference vector, whereas for Length 4 they might try to place the object to ¼ the distance. The children might use a similar strategy in Lengths 2 and 1, except that they may see the reconstructed vector as four times the reference vector in Length 2 and the double in Length 1. To test for this possibility, we proceeded to run a set of one-sample t-tests assuming perfect implementation of this strategy across all four lengths. Thus, the ideal vector to reconstruct would be 28.8 cm. for Length 4, 46.8 for Length 3, 100.8 for Length 2 and 115.2 for Length 1. We tested this model for both the Linear and the Right configurations using the original distance scores of the reconstructed vector. The results of the t-test

show that for Lengths 4 and 3, the 4-year-old children's responses did not differ significantly from the ideal vector (28.8 and 46.8, respectively) in both the Linear and the Right configurations (all ps. < .01). In contrast, for all other age groups, there were significant differences (all ps. > .1). In looking at Lengths 2 and 1, all configurations across the 4 age groups differ statistically from the ideal vectors (all ps. > .1). These results suggest that in the two shortest lengths, 4-year-olds used relative distances to scale the reconstructed vector. To better understand how children scaled the two longest vectors, we tested the preservation of the ordinal relations against chance and found that only 4- and 6-year-olds did not differ statistically from the random order in both the Linear and the Right configurations (Binomial test, all ps. > .05, two-tailed).

Why do 4-year-old children exhibit such divergent patterns of performance between the two shortest and two longest vectors? One possibility is that when the reconstructed vector is shorter than the reference vector, this array could be assimilated to an enclosed space situation. Previous studies have shown that children this age are capable of using a relative scaling strategy when the map-reading task takes place in a real 3-D layout (e.g., in a sand-box; 19). That is, children represent perceptual proportions, such as when the target object is half the distance between two landmarks, or one third the distance [19]. Moreover, children of this age also showed a bias to encode the object's spatial extent relative to a standard (e.g., a container) rather than encoding the object's absolute length [27]. In contrast, the longest vectors cannot be assimilated to an enclosed space situation because the reconstructed vector is longer than the reference vector, and thus children are not able to use a relative scaling strategy (e.g., double the size). Although the current results suggest that only 4-year-olds use relative distances to solve the scaling task, further studies should collect positive evidence for this claim by varying the scale factor [24, 25]. By this manipulation we could compare more directly the type of strategies that children of different ages use when scaling a configuration of objects. Beyond this matter, the current set of results concur with previous findings in showing that children between 4 and 8 years of age undergo substantial development in their scaling abilities [7, 8].

The results for the representation and preservation of angle showed two major transitions, between 4 and 6 years of age and between 8-year-olds and adults. The analysis of the main effect of age showed significant differences only in these two developmental periods. The results also showed that the significant difference between 4 and 6 years of age was mainly driven by an increase in precision in the Obtuse configuration. However, the current data also indicate a more progressive change in participants' success in representing angle across development. The analyses of the rate of success show an upward trend with 52%, 70%, 84% and 98% for 4-year-olds, 6-year-olds, 8-year-olds, and adults, respectively. This trend was also more visible for the Obtuse configuration. Thus, unlike the development of distance, the children's ability to represent and preserve angle information improves steadily from 4 years of age to adulthood. Additionally, the fact that a number of 4-year-old children display sensitivity to angle information indicates that a proportion of even younger children may succeed at representing angle in the current completion task.

The pattern of results for angle shows that both configuration variables, namely, Configuration type and Relative vector length, had an effect only for the youngest age group. For these children, preserving angle in the Acute configuration was easier than preserving angle in the Obtuse configuration. In fact, in the shortest length of the Obtuse configuration, only the 37% of the children put the toy relatively close to the ideal angle of 45˚. This result contrast to the 4-year-old children's performance in the longest vector of the Acute configuration where 76% of the children put the toy close to the ideal angle. These findings reveal that the 4-year-old children's ability to represent and preserve angles in scaling tasks is restricted to certain geometrical configurations. The low rate of success in the Obtuse configurations indicates that

young children have difficulties encoding and preserving the object's angular position relative to the configuration's intrinsic axes of reference. In contrast, the 4-year-old's high rate of success with the longest vector of the Acute configuration could be the result of this array being the most similar to a prototypical triangle shape [28]. Young children may exploit this similarity to better encode the geometrical information in the map and scale the vector. Therefore, until 6 years of age, most of the children begin to exhibit an abstract representation of angle encompassing diverse geometrical configurations. These results are in the line with conclusions of previous studies showing that preschool-age children have a relatively weak representation of angles in map-reading tasks [14, 15].

The current results for distance and angle differ from the previous findings reported by Uttal [21]. Although in our study 4-year-olds exhibited inferior performance than the older children, they displayed good performance in some arrays. This pattern indicates that complex abilities to scale distances and preserve angle are already present in preschool-age children when using a task focused on object-to-object scenarios that could be more cognitively demanding than scaling distances and preserve angle in enclosed spaces. However, our study concurs with Uttal [21] in demonstrating that young children's scaling performance varies to a larger extent compared to older children's performance as a function of specific configuration variables.

We believe that several cognitive factors could be driving the sharp developmental change between 4 and 6 years of age in both angle and distance representation. For instance, 6-year-olds may implement a particular scaling strategy in a more efficient way than younger children [8]. Another possibility is that this development might be related to some changes in the children's spatial memory skills. Previous studies suggest that between 3 and 5 years of age the ability to encode locations relative to other objects and their intrinsic reference frame undergoes a substantial development [29, 30]. Thus, in the current research the 4-year-old children may have some trouble in encoding the object's location relative to the intrinsic reference frame. However, the fact that the 4-year-olds succeeded in some configurations demonstrates an emerging ability to encode the objects' location in object-to-object scenarios.

The findings of the representation and use of orientation in the current completion task show a different pattern of results for the vertical and lateral orientation. In contrast to adults, in the lateral right-left orientation, all 4- and 6-year-old children failed to represent this geometrical property. This result is more striking since we did not vary the orientation of the target object in the map across configurations. Interestingly, 8-year-olds showed a mixed pattern of results, choosing the correct side only in three linear arrays. These results are overall consistent with previous findings. Thus, for example, in a map-reading task, Spelke, et al. [15] found that 5 and 6-year-olds failed to use orientation information to distinguish an array from its mirror image. Similarly, by using a deviant detection paradigm, Dehaene, Izard, Pica and Spelke [31] found that children as old as 11 years of age failed to detect the "odd" figure when orientation was the only distinctive geometrical feature. However, the current set of results shows that 8 years could be the age from which children begin to develop sensitivity to orientation in the simplest geometrical configurations, such as a linear array. In contrast to these findings, all children succeeded in representing the vertical up-down orientation across all the arrays, showing that this polarity is correctly represented from an early age [32].

Overall, the current research shows three important features of how the children's use of geometric information in a scaling task changes along development. First, the ability to utilize information about angles and distances in an abstract geometric map develops mainly in two periods of time: between 4 and 6 years of age and between 8 years of age and adulthood. Particularly, the representation of distance shows a substantial improvement between 4 and 6 years of age, while the ability to preserve angle information seems to develop more gradually than

distance information along childhood. Second, the ability to preserve orientation information in a scaling task develops late, beyond 8 years of age, although some important progress occurs between 6 and 8 years of age in simple configurations. Third, 4-year-olds are able to succeed in both preserving angle and scaling distances when reading maps with some particular geometrical configurations. Children of this age perform well when scaling vectors shorter than the reference vector and when preserving angles in Acute configurations. Therefore, this study shows that young children are able to exploit an intrinsic system of reference alone without cues of extended surfaces to scale a geometrical configuration.

An open question in the current research is related to the type of scaling strategies that participants use in the completion map reading task. The pattern of results suggest that the youngest age group of children seemed to utilize a relative strategy with some configurations in contrast to older participants. If this hypothesis is correct, it would show a qualitative developmental change between 4 and 6 years of age in the use of geometric information to scale maps. This would be in sharp contrast to the reported developmental change between 8 years of age and adulthood, which appears more grounded in an increase in scaling precision. Future research should study the type of strategies that children use to scale object-to-object configurations by varying the scale factor and how they change over development.

## Supporting information

**S1 Dataset. Scaling scores.**
(XLSX)

## Acknowledgments

We thank the children and undergraduate students who agreed to participate in this study. We also thank Jose Fernando García, Silvana Cortés and Silvia Sandoval who collaborated with data collection.

## Author Contributions

**Conceptualization:** Yenny Otálora, Hernando Taborda-Osorio.

**Data curation:** Yenny Otálora, Hernando Taborda-Osorio.

**Formal analysis:** Yenny Otálora, Hernando Taborda-Osorio.

**Funding acquisition:** Yenny Otálora, Hernando Taborda-Osorio.

**Investigation:** Yenny Otálora, Hernando Taborda-Osorio.

**Methodology:** Yenny Otálora, Hernando Taborda-Osorio.

**Project administration:** Yenny Otálora.

**Writing – original draft:** Yenny Otálora, Hernando Taborda-Osorio.

**Writing – review & editing:** Yenny Otálora, Hernando Taborda-Osorio.

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
