## [Decision Letter · Decision Letter 0]

8 Jun 2020

PONE-D-20-10910

Developmental differences in children’s and adults’ use of geometric information in map-reading tasks

PLOS ONE

Dear Dr. Otálora,

Thank you for submitting your manuscript to PLOS ONE. After careful consideration, we feel that it has merit but does not fully meet PLOS ONE’s publication criteria as it currently stands. Therefore, we invite you to submit a revised version of the manuscript that addresses the points raised during the review process.

We look forward to receiving your revised manuscript.

Kind regards,

David Henry Uttal, PhD

Academic Editor

PLOS ONE

Additional Editor Comments:

Overall, this is a good paper. The research methods and analyses are sound, and the paper may make an important contribution to the literature on spatial scaling. The specific investigation of what children of different ages encode and scale in a mapping task sheds new light on the development of these abilities.

The reviewers offer very specific and helpful comments, and you should address all of their concerns in your response. In general, both call for substantially more detail regarding the methods. I agree with these suggestions, and at the same time, I'd like to consider making clearer how the methods you are using.

I now turn some specific concerns from my own reading of the manuscript:

a) The writing is generally quite good throughout. There are a few situations, however, in which I thought a phrase could be said more clearly or with fewer words

Line 164: "...were variables affecting children' performance". Perhaps simply say, "The absolute size and symmetry of the configuration affect children's performance." or "The variables of interest were absolute size and symmetry."

Line 165, "By address the effect of some variables..." Say specifically which variables.

Line 176, "to call the quantity", perhaps "It is more precise to label this quantity as a vector"

Line 281, Insert "in" before "representing"

Lines 410-413: Could this sentence be stated more succinctly?

Line 457-458, "These children failed to reach a significant difference", change to something like, "The results did not differ from chance in the 3 longest arrays."

LIne 478, "From early ages" could simply be "young children"

b) I was confused by both the placement and organization of the tables. I was not sure why the Table is included in the text but the figures are not. Please follow PLOS1 guidelines for the placement of tables and figures. In addition, the figures are too small and quite hard to read at times. The entrees in the legends are very small, and readers will have difficulty mapping the legend entrees to the lines in the graph.

c) I found it somewhat difficult to interpret the age differences between the children and adults. You refer several times to developments happening between the ages of 8 and adulthood. However, because you have no age groups between this range, it seems very non-specific. I know you already have several age groups, and testing beyond age 8 was not your focus. Could you say simply, "Beyond age 8" or something like that to make it simpler?

d) Your interpretation of the findings in the Discussion is interesting, but there is a great deal of speculation and relatively little that is directly tied to the results. I wondered if you could focus more on what the results say and perhaps less on speculative interpretations. Of course, some speculation is justified if it is marked as such

e) I believe there are several highly relevant publications by Moira Dillon that could enhance your literature review and perhaps provide a more specific motivation for the work.,

f) Finally, let me address one comment from Reviewer 1. She/he wondered whether the paper makes a significant contribution to the literature. I note that the magnitude of the contribution is explicitly not a criteria for this journal, so you do not have to address that comment directly. However, I do think that this reviewer's concern reflects an aspect of the manuscript that you should address: Sometimes the relation between the research questions and the methods is not clear. You could be more explicit as to why you included a specific methodological manipulation in terms of the questions that you are asking.

Overall, I think this paper has potential and encourage you to complete the necessary revisions.

2. Please provide additional details regarding participant consent.

In the ethics statement in the Methods and online submission information, please ensure that you have specified whether consent was informed.

Reviewers' comments:

Reviewer's Responses to Questions

**Comments to the Author**

1. Is the manuscript technically sound, and do the data support the conclusions?

Reviewer #1: Partly

Reviewer #2: Yes

2. Has the statistical analysis been performed appropriately and rigorously? 

Reviewer #1: Yes

Reviewer #2: Yes

3. Have the authors made all data underlying the findings in their manuscript fully available?

Reviewer #1: Yes

Reviewer #2: Yes

4. Is the manuscript presented in an intelligible fashion and written in standard English?

Reviewer #1: Yes

Reviewer #2: Yes

5. Review Comments to the Author

Reviewer #1: The questions addressed in the manuscript concern developmental changes in the use of geometric information on mapping tasks. The introduction provides a comprehensive overview of relevant work, pointing out gaps in the current understanding of geometric mapping in children. In particular, the authors indicate that it is not clear how the three key geometric dimensions (distance, angle, and orientation) are weighted by children in mapping tasks and whether certain dimensions are more challenging than others (e.g., whether preserving angles in mapping is harder than scaling distance). The experimental method includes a task, in which participants must find a location in a larger space corresponding to the target location on a smaller picture; the task can be solved by taking into account all three types of geometric information.

While I see multiple strengths in the paper (interesting problem area, insightful literature analysis, clever experimental task, in-depth analysis), I’m not sure it represents a substantial advancement in current knowledge. In particular, I don’t think it fills the identified gaps in the literature, in particular, the issues of relative weighting and relative difficulty of different types of geometric information. There are a lot of specific findings reported, but the main take-away message is that children, even as young as 4 y. o., have some ability to use geometric cues in mapping and that this ability undergoes substantial development. The multiple nuanced findings don’t seem to add up to a general picture beyond what has been already established.

In addition to this concern, I have several further comments listed below.

1. I was curious as two why the authors chose two variables – configuration type and relative vector length – to examine the conditions that affect children’s performance on their task. The description of these two variables was very clear but there was no relevant theoretical rationale or predictions. This made it harder to interpret many of the findings. For example, distance analysis showed “significant differences between Linear and Right configurations only for Length 2, with more errors in the linear configuration.” It is hard to say if it is a meaningful finding – whether it is consistent with the authors’ expectations and, therefore, how it should be interpreted.

2. Related to (1), it is not clear why the authors reported the results of their pair-wise analysis as planned comparisons. Typically, planned comparisons are theoretically based, but I don’t think the authors predicted, for example, that the difference between linear and right configurations will be found only for Length 2. Unless there is a strong rationale for the use of planned comparisons throughout Results, I think the analyses should be re-run, controlling for the overall error in multiple comparisons. This may turn some of the significant findings into non-significant.

3. In my opinion, it was unfortunate that the distance analysis was limited to two configurations: Linear and Right triangle. I understand that the other two configurations did not include all 4 lengths, but perhaps the authors could compare children’s accuracy with the two overlapping lengths that were used across all four configurations. This could show whether the accuracy (and, by inference, difficulty) of distance scaling varies depending on the need to preserve angular information. In the current version, the two configurations represented “prototypical” angles: 90 and 180 degrees (linear array). With these two configurations, no differences were found in distance scaling. But the findings could be different if children had to scale distance while preserving a less prototypical angle (e.g., 135 degrees). It would be interesting to see if the need to preserve this angular information made scaling more challenging and whether this varied with age.

4. One of the interesting findings is that accuracy of distance scaling increased substantially between 8 and 20 years of age. The authors pointed out that this finding is different from prior reports, and that the difference could be due to the fact that prior studies involved distance scaling in the enclosed space, whereas the current study required coding distances between objects. I would like to see a more extensive discussion of this finding. One possible explanation could be the difficulty of distance scaling based on part/part vs. part/whole relations. Many of the prior studies allowed children to code the target location relative to the frame, which could be done by considering a part/whole relation (e.g., a distance between the edge of the box and the target location relative to the length of the whole box). In the present study, all configurations except for the linear one required considering a different type of relation – part/part (i.e., reference vector/reconstructed vector). This is, of course, only one possible explanation and the authors might offer other interpretations.

5. The discussion of 4-y.o. performance on p. 24 is somewhat confusing because of the terminology used. The authors suggest that the youngest children may be utilizing relative distance coding (unlike other age groups). All the age groups must be utilizing some sort of relative distance coding on a mapping task. The phenomenon discussed by the author with respect to 4-y.o. sounds similar to “categorical distance coding” whereby distances are coded in terms of halves and quarters rather than a more fine-grained coding.

Reviewer #2: This is a clear manuscript which is well written. The authors investigated the map reading abilities of young children and adults in various conditions which involved children (and adults) extrapolating information they had seen on a map to a real world space that it represented. The tasks involved placing one object in a space in relation to two other (already present) objects. The authors measured participants’ performance in terms of distance accuracy and angular accuracy. The results show an age related improvement in the ability to perform the tasks accurately.

The introduction is clear and provides an appropriate background for the study.

The procedure and method are clear but need a little more explanation and justification. The children were compared to adults, but the adults were psychology students and presumably had a higher mean IQ than the mean of the child age groups. This may not matter, but a more representative sample of adults would have been better. Did any of the adults have special experience with maps (e.g. from cartography classes, orienteering, map making, or map training, etc.)? Had the children had any map reading or map using classes?

The design of the materials is clear, and was appropriate for the tasks in the experiment. [But more emphasis might be given in the discussion to point out these were very denuded maps, and any findings might not transfer to even slightly more sophisticated maps or spatial representations.]

The procedure needs a lot more detail. Please include a plan of where the experimenter and the participant were when they looked at the map on the table, and where the two sheep were in relation to the participant. Presumably the participants could not see the sheep on the ground at the same time as they were looking at the map (so the task involved memory)?

The authors showed the participants the sheep on the ground and then showed the participants the map. How was this done? By looking at the map, then looking at the sheep on the ground, then looking at the map again?? How many times were participants allowed to look between the ground and the map, and for how long? Or could participants not see the ground at all while they were looking for up to 5 seconds at the map?

What was the orientation of the map on the table in relation to the sheep on the ground (i.e. was the map aligned with the space?). If the map was aligned with the space the participants were, in effect, having to mentally rotate the information on the map to match the space. If the map was aligned with the space but the participants were looking at it ‘upside down’ mental rotaton might also be involved. This points are not clear in the procedure and this is why it is essential to include a plan of the space/participants/table and the orientation of the map.

There is a memory component – the participants did not have the map with them so they had to remember where the X was on the map when they turned to the space. There might also have been a metacognitive component (or several of these) – for example, did the adults spend longer looking at the map than did the children – assuming the adults knew they had to encode an angle or a distance and gave themselves time to this. In other words did the older children and adults look for the full 5 seconds and the younger children look for less time? Children may have thought they could encode the angle/distance in a glance. Did time looking at the map vary with the ‘difficulty’ of the task?

Why was a maximum of 5 seconds chosen for the time looking at the map? Might this have put the younger participants at a disadvantage?

There might have been a strategy component – how did participants turn from the map table? – did some even walk round the map table to get to the sheep? Or were they not allowed to? How did the researcher ‘turn around the child’ (p12)

The participants did 12 trials. Were these in a different random order for EACH participant (this is not clear on p12)? If the maps were in the same random order for all participants how might practice effects have influenced the results.

Participants received feedback for the two practice, but no feedback after that – I assume that without some feedback it was hard to keep the children, especially the 4 year olds, motivated to go on doing the task. I assume some of the younger children wanted to place the lion where they wanted to, rather than follow the instructions to place the lion like on the map. How were these sort of issues dealt with?

The results are clearly expressed.

The discussion is a summary of the results. But the discussion introduces new analyses (p24). All data and analyses should be in the results section.

The rest of the discussion is a good summary of the present results. But the discussion needs to take into account other factors (some mentioned above) that might have effected participants’ performance. Factors that can be dismissed with good reasons, or factors that could have affected all ages groups equally should be considered. Factors like greater memory load once the mapswas covered and which might have affected the younger children most need to discussed, even if they are to be discounted. The authors need to demonstrate that all their results are due to map reading/reasoning (as the authors imply) and are not an artefact of the procedure (e.g. older participants’ longer time looking at the map; or older participants’ better mental rotation skills, and so on).

Some decimal points seem to be represented by a full stop, and some by a comma - these need checking for consistency

6. PLOS authors have the option to publish the peer review history of their article (what does this mean?). If published, this will include your full peer review and any attached files.

Reviewer #1: No

Reviewer #2: No

---

## [Author Response · Author response to Decision Letter 0]

20 Jul 2020

Dear Dr. Otálora,

Thank you for submitting your manuscript to PLOS ONE. After careful consideration, we feel that it has merit but does not fully meet PLOS ONE’s publication criteria as it currently stands. Therefore, we invite you to submit a revised version of the manuscript that addresses the points raised during the review process.

We look forward to receiving your revised manuscript.

Kind regards,

David Henry Uttal, PhD

Academic Editor

PLOS ONE

Additional Editor Comments:

Overall, this is a good paper. The research methods and analyses are sound, and the paper may make an important contribution to the literature on spatial scaling. The specific investigation of what children of different ages encode and scale in a mapping task sheds new light on the development of these abilities.

The reviewers offer very specific and helpful comments, and you should address all of their concerns in your response. In general, both call for substantially more detail regarding the methods. I agree with these suggestions, and at the same time, I'd like to consider making clearer how the methods you are using.

I now turn some specific concerns from my own reading of the manuscript:

a) The writing is generally quite good throughout. There are a few situations, however, in which I thought a phrase could be said more clearly or with fewer words

Line 164: "...were variables affecting children' performance". Perhaps simply say, "The absolute size and symmetry of the configuration affect children's performance." or "The variables of interest were absolute size and symmetry."

Answer: We rewrote the paragraph (Lines 173-187) and we did not mention these variables.

Line 165, "By address the effect of some variables..." Say specifically which variables.

Answer: It was stated as follows: “By addressing the effect of these variables” (Line 208). We used the deictic “these” because we named the variables within the sentence located before.

Line 176, "to call the quantity", perhaps "It is more precise to label this quantity as a vector"

Answer: It was stated as follows: “it is more precise to label this quantity as a “vector”.” (Line 181)

Line 281, Insert "in" before "representing"

Answer: We inserted “in” before “representing” in the first paragraph of the data collection section (Lines 334, 336 and 338)

Lines 410-413: Could this sentence be stated more succinctly?

Answer: It was stated as follows: “Overall, the prior findings reveal that Configuration Type and Vector Length just affected the 4-year-old children’s scaling ability, as their performance differed when they tried to scale Acute versus Obtuse configurations and Short versus Long vectors.” (Lines 460-462)

Line 457-458, "These children failed to reach a significant difference", change to something like, "The results did not differ from chance in the 3 longest arrays."

Answer: It was stated as follows: “These results did not differ from chance across all configurations except in the 3 longest linear arrays (all ps. < .05)” (Lines 508-509)

Line 478, "From early ages" could simply be "young children"

Answer: It was stated as follows: “Young children” (Line 672)

b) I was confused by both the placement and organization of the tables. I was not sure why the Table is included in the text but the figures are not. Please follow PLOS1 guidelines for the placement of tables and figures.

We submitted the figures as individual files because the submission guidelines for Figures establish: “Do not include figures in the main manuscript file. Each figure must be prepared and submitted as an individual file.”

Answer: We revised the placement and organization of the Figures and left the citations and captions of these Figures in the same place because the submission guidelines for Figures establish: “Cite figures in ascending numeric order at first appearance in the manuscript file. Figure captions must be inserted in the text of the manuscript, immediately following the paragraph in which the figure is first cited (read order). Do not include captions as part of the figure files themselves or submit them in a separate document.”

It is important to point out that we cited Figure 1 twice within two different paragraphs, first within the subsection Materials to illustrate the elements on the map, and second within the subsection Design to illustrate the task variables. However, Figure 1’s captions were placed between these two paragraphs immediately following the paragraph in which the figure is first cited. We just moved the second citation of Figure 1 to the end of the sentence to avoid splitting this sentence, while preserving the meaning of the citation.

Moreover, following the ‘PLOS ONE manuscript body formatting guidelines’ we also confirmed that we should leave one space between the paragraph and the figure captions.

We placed the table within the manuscript file directly after the paragraph in which it is first cited because the submission guidelines for Tables establish: “Cite tables in ascending numeric order upon first appearance in the manuscript file. Place each table in your manuscript file directly after the paragraph in which it is first cited (read order). Do not submit your tables in separate files. Tables require a label (e.g., “Table 1”) and brief descriptive title to be placed above the table. Place legends, footnotes, and other text below the table.”

In addition, the figures are too small and quite hard to read at times. The entrees in the legends are very small, and readers will have difficulty mapping the legend entrees to the lines in the graph.

Answer: Thank you for noticing this. We have corrected all graphs now.

c) I found it somewhat difficult to interpret the age differences between the children and adults. You refer several times to developments happening between the ages of 8 and adulthood. However, because you have no age groups between this range, it seems very non-specific. I know you already have several age groups, and testing beyond age 8 was not your focus. Could you say simply, "Beyond age 8" or something like that to make it simpler?

Answer: We are aware of this issue indeed. In the manuscript we refer to differences between 8-year-olds and adults because those were the groups we tested and because our interest was to compare the children’s performance with adults to look for possible differences. However, we are aware that from this comparison we cannot be certain as to when exactly this change occurs. Consequently, we took advantage of one comment from Reviewer 2 (asking for an explanation about differences in performance between children and adults) and added some lines (page 25, lines 531-534) addressing this issue. We hope this comment addresses your main concern.

d) Your interpretation of the findings in the Discussion is interesting, but there is a great deal of speculation and relatively little that is directly tied to the results. I wondered if you could focus more on what the results say and perhaps less on speculative interpretations. Of course, some speculation is justified if it is marked as such

Answer: In answering some of the reviewers’ comments we had to expand on additional interpretations of the main findings. We tried to reach a balance in the Discussion section between presenting the results and presenting our interpretations. We decided to keep these interpretations in the manuscript as they seek to clarify the findings in more relevant theoretical terms. However, we tried to mark these interpretations as speculations.

e) I believe there are several highly relevant publications by Moira Dillon that could enhance your literature review and perhaps provide a more specific motivation for the work.,

Answer: Yes, we agree that the Dillon and Spelke’s work is highly relevant to our study as these authors investigate the children’s representational abilities they utilize when reading maps and how these abilities change over development. Accordingly, we wrote a paragraph (Fifth paragraph; Lines 94-103) in the Introduction section presenting how their research informs our own study.

f) Finally, let me address one comment from Reviewer 1. She/he wondered whether the paper makes a significant contribution to the literature. I note that the magnitude of the contribution is explicitly not a criteria for this journal, so you do not have to address that comment directly. However, I do think that this reviewer's concern reflects an aspect of the manuscript that you should address: Sometimes the relation between the research questions and the methods is not clear. You could be more explicit as to why you included a specific methodological manipulation in terms of the questions that you are asking.

Answer: We acknowledge that the connection between the research question and the methods was rather unclear in the original manuscript. We have now expanded the Current Study section substantially to address this issue (Fourth paragraph; Lines 196-213). We state here the reasons that lead us to manipulate the configuration variables that we used in the study. Regardless of why we manipulated some particular variables, our interest in using different configurations of objects was to try to uncover differences in the participants’ performance when scaling maps. We show that only 4-year-olds exhibited a clear configuration effect, however. We hope the information we added renders the connection between the research question and the method clearer now.

Overall, I think this paper has potential and encourage you to complete the necessary revisions.

2. Please provide additional details regarding participant consent.

In the ethics statement in the Methods and online submission information, please ensure that you have specified whether consent was informed.

Answer: We complemented the information regarding participant consent within the Participants section, and put all information regarding the ethics procedures of the study together in one paragraph (Line 224-230). There, we clarified that the consent was informed.

Answer: We included captions for our Supporting Information files at the end of the manuscript (Lines 772-773), based on the Supporting Information guidelines. We did not include any in-text citations of Supporting Information files.

Reviewers' comments:

Reviewer's Responses to Questions

Comments to the Author

1. Is the manuscript technically sound, and do the data support the conclusions?

Reviewer #1: Partly

Reviewer #2: Yes

2. Has the statistical analysis been performed appropriately and rigorously?

Reviewer #1: Yes

Reviewer #2: Yes

3. Have the authors made all data underlying the findings in their manuscript fully available?

Reviewer #1: Yes

Reviewer #2: Yes

4. Is the manuscript presented in an intelligible fashion and written in standard English?

Reviewer #1: Yes

Reviewer #2: Yes

5. Review Comments to the Author

Reviewer #1: The questions addressed in the manuscript concern developmental changes in the use of geometric information on mapping tasks. The introduction provides a comprehensive overview of relevant work, pointing out gaps in the current understanding of geometric mapping in children. In particular, the authors indicate that it is not clear how the three key geometric dimensions (distance, angle, and orientation) are weighted by children in mapping tasks and whether certain dimensions are more challenging than others (e.g., whether preserving angles in mapping is harder than scaling distance). The experimental method includes a task, in which participants must find a location in a larger space corresponding to the target location on a smaller picture; the task can be solved by taking into account all three types of geometric information.

While I see multiple strengths in the paper (interesting problem area, insightful literature analysis, clever experimental task, in-depth analysis), I’m not sure it represents a substantial advancement in current knowledge. In particular, I don’t think it fills the identified gaps in the literature, in particular, the issues of relative weighting and relative difficulty of different types of geometric information. There are a lot of specific findings reported, but the main take-away message is that children, even as young as 4 y. o., have some ability to use geometric cues in mapping and that this ability undergoes substantial development. The multiple nuanced findings don’t seem to add up to a general picture beyond what has been already established.

Answer: We appreciate that you point out both the strengths and the weaknesses of our manuscript. We believe this research represents an important advancement because it explores two issues that haven’t been studied before systematically: the proficiency in the use of all three dimensions of geometric information in a scaling task and how this proficiency interacts with specific configuration variables. Thanks to the use of an object-to-object scenario we can show similarities and differences in the use of the three Euclidean properties across development. The main purpose of using both configuration variables (relative length and configuration type) was to determine whether these developmental trajectories change when we look at different configurations of objects. We show that this is partially true. Only 4-year-olds exhibited a clear configuration effect, but this was enough to show that the question about the origins of the scaling ability depends in part on the type of task or configuration implemented. Thus, for instance, young children show clear opposite performances in short and large vectors, very low for the latter and very high for the former. We now try to clarify the rationale for using these variables in the fourth paragraph of the Current Study section (Lines 196-213).

In addition to this concern, I have several further comments listed below.

1. I was curious as two why the authors chose two variables – configuration type and relative vector length – to examine the conditions that affect children’s performance on their task. The description of these two variables was very clear but there was no relevant theoretical rationale or predictions. This made it harder to interpret many of the findings. For example, distance analysis showed “significant differences between Linear and Right configurations only for Length 2, with more errors in the linear configuration.” It is hard to say if it is a meaningful finding – whether it is consistent with the authors’ expectations and, therefore, how it should be interpreted.

Answer: We agree that the rationale for choosing both configuration variables was unclear in the manuscript. We now have presented a longer justification for choosing both variables in the fourth paragraph of the Current Study section (Lines 196-213). We chose these variables because they both are constitutive dimensions of a configuration of objects in any reconstruction task and they allowed us to explore how the participants’ performance changes across variations in distance and angle, which are the defining dimensions of the Euclidean space. Overall, we believe that the manipulation of these configuration variables allows us to uncover some differences in the children’s performance in map-reading tasks. As we say on page 9, our interest in this research was not only to present a developmental trajectory but also to explore the task conditions under which children display a better performance.

2. Related to (1), it is not clear why the authors reported the results of their pair-wise analysis as planned comparisons. Typically, planned comparisons are theoretically based, but I don’t think the authors predicted, for example, that the difference between linear and right configurations will be found only for Length 2. Unless there is a strong rationale for the use of planned comparisons throughout Results, I think the analyses should be re-run, controlling for the overall error in multiple comparisons. This may turn some of the significant findings into non-significant.

Answer: Thank you very much for noticing this. We definitely agree with your remark. We already changed the pair-wise analysis from planned comparisons to post hoc comparisons (bonferonni test). Originally, we had controlled the Type I error for multiple comparisons any way, so except for one analysis (page 17, line 385) all others show the same overall pattern of results.

3. In my opinion, it was unfortunate that the distance analysis was limited to two configurations: Linear and Right triangle. I understand that the other two configurations did not include all 4 lengths, but perhaps the authors could compare children’s accuracy with the two overlapping lengths that were used across all four configurations. This could show whether the accuracy (and, by inference, difficulty) of distance scaling varies depending on the need to preserve angular information. In the current version, the two configurations represented “prototypical” angles: 90 and 180 degrees (linear array). With these two configurations, no differences were found in distance scaling. But the findings could be different if children had to scale distance while preserving a less prototypical angle (e.g., 135 degrees). It would be interesting to see if the need to preserve this angular information made scaling more challenging and whether this varied with age.

Answer: This is indeed an interesting comparison. We ran the analysis in the way you suggest and we did not find significant differences in distance across all four configurations, meaning that scaling was not more challenging when preserving angle in any age group. We don’t include this analysis in the manuscript as it is not part of the intendent plan of analyses. But if you think otherwise, we are willing to include this analysis in the results section, even if it’s negative it could be interesting to report.

4. One of the interesting findings is that accuracy of distance scaling increased substantially between 8 and 20 years of age. The authors pointed out that this finding is different from prior reports, and that the difference could be due to the fact that prior studies involved distance scaling in the enclosed space, whereas the current study required coding distances between objects. I would like to see a more extensive discussion of this finding. One possible explanation could be the difficulty of distance scaling based on part/part vs. part/whole relations. Many of the prior studies allowed children to code the target location relative to the frame, which could be done by considering a part/whole relation (e.g., a distance between the edge of the box and the target location relative to the length of the whole box). In the present study, all configurations except for the linear one required considering a different type of relation – part/part (i.e., reference vector/reconstructed vector). This is, of course, only one possible explanation and the authors might offer other interpretations.

Answer: The difference in performance between 8-year-olds and adults is indeed very salient in our results and different from previous studies. We believe this difference could be related to a more demanding metric coding when remembering the objects’ position in open spaces compared to the metric coding in enclosed spaces, this is something you bring up in the next point. In enclosed spaces, like a rectangle (see Sandberg, Huttenlocher, & Newcombe, 1996), participants can use “spatial categories” to encode the objects’ position which we believe could be harder for children to represent in open spaces. Another possibility has to do with the implementation of a “mental transformation strategy” which researchers believe participants utilize when scaling distances in enclosed spaces. The use of this strategy could be harder in open spaces because of the lack of visible boundaries that helps people to mentally expand the space. We now have expanded on this issue in the second paragraph of the Discussion section (Lines 526-544).

5. The discussion of 4-y.o. performance on p. 24 is somewhat confusing because of the terminology used. The authors suggest that the youngest children may be utilizing relative distance coding (unlike other age groups). All the age groups must be utilizing some sort of relative distance coding on a mapping task. The phenomenon discussed by the author with respect to 4-y.o. sounds similar to “categorical distance coding” whereby distances are coded in terms of halves and quarters rather than a more fine-grained coding.

Answer: Thank you for your comment, I see the reason of your confusion. On this page we are referring to a “relative scaling strategy”, not relative or categorical coding. We now clarify what we mean to say by this kind of strategy on page 27 (line 582). The issue about whether children and adults use this or other strategies when scaling distances has been a matter of much discussion in the scaling literature. Some studies show evidence for the use of the relative scaling strategy, something like perceptual proportions which are constant regardless of the size of the configuration, while other studies show evidence for the use of other strategies (e.g. mental transformation). We believe our evidence is more consistent with the use of the relative scaling strategy in some configurations, but we are clear in the manuscript as to we don’t have strong evidence for this, that is why we propose this only as a possible interpretation of our results worth exploring in future studies. Of course, we agree children may be using some sort of categorical coding, but they might not be using this as a strategy to scale distances.

Reviewer #2: This is a clear manuscript which is well written. The authors investigated the map reading abilities of young children and adults in various conditions which involved children (and adults) extrapolating information they had seen on a map to a real world space that it represented. The tasks involved placing one object in a space in relation to two other (already present) objects. The authors measured participants’ performance in terms of distance accuracy and angular accuracy. The results show an age related improvement in the ability to perform the tasks accurately.

The introduction is clear and provides an appropriate background for the study.

The procedure and method are clear but need a little more explanation and justification. The children were compared to adults, but the adults were psychology students and presumably had a higher mean IQ than the mean of the child age groups. This may not matter, but a more representative sample of adults would have been better. Did any of the adults have special experience with maps (e.g. from cartography classes, orienteering, map making, or map training, etc.)? Had the children had any map reading or map using classes?

Answer: We agree that a more representative sample of adults would have been better, but psychology students are the most accessible population to run this type of studies and this is true for the majority of studies in psychology. Moreover, a priori we don’t see fundamental reasons as to why psychology students would have better spatial skills compared to the rest of the population. Students in psychology don’t have special experience with maps; any of their curriculum courses included teaching about this type of representation. The children did not participate in school courses using maps. Moreover, geometry classes in Colombia typically begin in 4th grade and do not involve map-reading activities.

The design of the materials is clear, and was appropriate for the tasks in the experiment. [But more emphasis might be given in the discussion to point out these were very denuded maps, and any findings might not transfer to even slightly more sophisticated maps or spatial representations.]

Answer: We specified that the maps utilized in the study were purely geometric maps (Line 160 and line 233). We rewrote a part of the first paragraph in the Current Study section to justify the use of this type of map in the study: “The use of purely geometric maps devoid of any landmark allows us to examine how the participants extract and represent the three Euclidian properties from the configuration of objects: distance, angle and orientation. Moreover, this type of completion task allows us to investigate the developmental trajectory for each of these three Euclidian dimensions that participants have to represent when solving the task.” (Lines 166-171).

The procedure needs a lot more detail. Please include a plan of where the experimenter and the participant were when they looked at the map on the table, and where the two sheep were in relation to the participant. Presumably the participants could not see the sheep on the ground at the same time as they were looking at the map (so the task involved memory)?

Answer: We agree with this comment and others regarding the procedure. Therefore, we complemented the three paragraphs of the Procedure section with more detailed information.

In the first paragraph of this section, where we described the general procedure, we specified that: “Each participant stood in front of the table and was able to observe the map from above. The researcher was positioned directly on the left side of the participant in front of the table and was able to observe the map from above and point out the target location on the map. The three-dimensional layout was located behind both of them 3 mt. apart, where only the two sheep were already placed (Fig 2). Each participant was asked to look at the map, and then go to the three-dimensional space and locate the lion in the place represented by an X on the map” (Lines 275-281). Here we cited a new figure (Fig 2), which illustrates the experimental set up. It is correct that the participants could not see the sheep on the ground at the same time as they were looking at the map, so the task involved memory. Therefore, in this paragraph we also clarified that “The map and the three-dimensional layout were never visible simultaneously for the participant. After looking at the map, the participant was asked to rotate 180º and walk to the place where he/she decided to locate the lion.” (Lines 281-284).

The authors showed the participants the sheep on the ground and then showed the participants the map. How was this done? By looking at the map, then looking at the sheep on the ground, then looking at the map again?? How many times were participants allowed to look between the ground and the map, and for how long? Or could participants not see the ground at all while they were looking for up to 5 seconds at the map? 

There might have been a strategy component – how did participants turn from the map table? – did some even walk round the map table to get to the sheep? Or were they not allowed to? How did the researcher ‘turn around the child’ (p12)

Answer: To answer these two comments, we introduced more detailed information to clarify the specific procedures during both the practice phase and the test phase sections. It is because there were differences between both phases regarding how the experimenter presented the task. 

During the practice phase, a large prompt to familiarize the participants with the task was utilized. When the experimenter told the child that the picture (map) was the drawing of the space behind them and presented the two sheep, she showed the 3D-space to the child. When we described this, we specified how she turned the child around: “(holding the child’s shoulders and turning him/her around 180º in clockwise direction, showing him/her where the sheep were positioned and again turning the child around in clockwise direction to look at the map).” (Lines 295-298). As we specified before, “The map and the three-dimensional layout were never visible simultaneously for the participant.” (Lines 281-282). At the end of the prompt we also clarified that: “The researcher gave the child 5 seconds to observe the map. Then, she took away the map and told the child “You can go now”. The researcher held the child’s shoulders and turned him/her around 180º in clockwise direction. Then she walked 1mt straightforward with the child and stood in front of the configuration of sheep, then she waited for the child to locate the lion. The child was not allowed to walk around the table or take any different direction to that instructed by the researcher.” (Lines 302-308).

During the test phase section, we had stated that a summarized prompt was utilized and now we clarified that “the researcher showed the map to the children just once at the end of the prompt” (Lines 316-317). And at the end of the prompt we clarified that: “The researcher gave the child 5 seconds to observe the map and took it away. The researcher held the child’ shoulders and turned him/her around 180º in clockwise direction. Then, the child walked 1mt straightforward as instructed, stood in front of the configuration of sheep and located the lion on the ground. The researcher waited for the child to locate the lion and located the next map on the table for a new trial.” (Lines 321-326).

We should stand out that during the practice phase the researcher showed two times the 3D-layout to the child for each practice trial. The first time, the child did not walk towards the 3D-layout. The researcher just turned the child around 180º to look at the 3D-space, and turned him/her around again to look at the map. The second time, the child walked towards the 3D-layout to solve the task and the researcher walked with the child. Both procedures were taken just to familiarize the child with the instructions and materials of the task and for that reason during the test trails the participants just turned around one time at the end of the prompt and walked alone towards the 3D-space to locate the lion in the target location. These procedures were piloted with participants from the different ages before collecting data. We found the procedures sufficient for the participants to understand the instructions of the task.

What was the orientation of the map on the table in relation to the sheep on the ground (i.e. was the map aligned with the space?). If the map was aligned with the space the participants were, in effect, having to mentally rotate the information on the map to match the space. If the map was aligned with the space but the participants were looking at it ‘upside down’ mental rotaton might also be involved. This points are not clear in the procedure and this is why it is essential to include a plan of the space/participants/table and the orientation of the map.

Answer: Regarding the orientation, we also clarified in the general procedure section that: “The orientation of the configuration of objects in the three-dimensional layout relative to the participant had the same orientation than the map relative to the participant” (Lines 284-286). This means that the task did not require mental rotation to match the information on the map with the 3D-space. Moreover, we now introduced a new Figure 2, which shows the orientation of the map and the orientation of the 3D-layout.

There is a memory component – the participants did not have the map with them so they had to remember where the X was on the map when they turned to the space. There might also have been a metacognitive component (or several of these) – for example, did the adults spend longer looking at the map than did the children – assuming the adults knew they had to encode an angle or a distance and gave themselves time to this. In other words, did the older children and adults look for the full 5 seconds and the younger children look for less time? Children may have thought they could encode the angle/distance in a glance. Did time looking at the map vary with the ‘difficulty’ of the task?

Why was a maximum of 5 seconds chosen for the time looking at the map? Might this have put the younger participants at a disadvantage?

Answer: We established the same time looking at the map at the end of the prompt for all the participants of the study, independently from the difficulty level of the task or the age of the participants. From the prior literature, we learnt that 2-3 seconds was enough time to look at the map for young children. During the pilot study, we tested the looking time and found that all the participants needed just about 3 seconds to look at the map, including the youngest children. After looking at the map during 3 seconds, these children turned around and walked to the 3D-space to locate the object in the target location. Therefore, for the test trials we left a wider fixed range of time (5 seconds) to allow for participants that potentially needed more time. Then, the youngest children were not at a disadvantage compared to the other participants. Moreover, we did not allow that participant leave the table before the 5 seconds. All the participants could explore the map during the 5 seconds.

The participants did 12 trials. Were these in a different random order for EACH participant (this is not clear on p12)? If the maps were in the same random order for all participants how might practice effects have influenced the results.

Answer: We clarified that “During the test phase all 12 maps were presented in random order for each participant”, in the third paragraph of the Procedure section (Lines: 326-327)

Participants received feedback for the two practice, but no feedback after that – I assume that without some feedback it was hard to keep the children, especially the 4 year olds, motivated to go on doing the task. I assume some of the younger children wanted to place the lion where they wanted to, rather than follow the instructions to place the lion like on the map. How were these sort of issues dealt with?

Answer: During the pilot study, we found that the 4 and 6-year-old children remained focused on the task and they didn’t do activities unrelated to the task. Moreover, the task was presented as a game. We also utilized a developmentally appropriated prompt that was clear for them. Overall, we noted that the task resulted engaging for the participants of all ages. We also piloted the number of maps the children would solve without getting tired or uninterested. We found that 12 maps were a good number to have the children engaged along the study and focused on the situation.

The results are clearly expressed.

The discussion is a summary of the results. But the discussion introduces new analyses (p24). All data and analyses should be in the results section.

Answer: We present these statistical results as a post hoc analysis, we didn’t plan to carry out these analyses in advance when we set up the design. These results suggest that young children are using a different scaling strategy than older participants, but we are clear as to say that we don’t have conclusive evidence for this hypothesis. Hence, we would like to keep this analysis in the current section of the manuscript as merely suggesting an interesting hypothesis.

The rest of the discussion is a good summary of the present results. But the discussion needs to take into account other factors (some mentioned above) that might have effected participants’ performance. Factors that can be dismissed with good reasons, or factors that could have affected all ages groups equally should be considered. Factors like greater memory load once the mapswas covered and which might have affected the younger children most need to discussed, even if they are to be discounted. The authors need to demonstrate that all their results are due to map reading/reasoning (as the authors imply) and are not an artefact of the procedure (e.g. older participants’ longer time looking at the map; or older participants’ better mental rotation skills, and so on).

Answer: We don’t believe the results are an artifact of older participants’ longer looking at the map as participants overall -children and adults- usually expended no more than 3 seconds looking at each map in the test trials. Of course, some small differences in looking time could be relevant in better encoding spatial locations, but the fact that even 4-year-olds succeeded in some trials shows that each participant understood the task and expended the time they needed to accomplish it. However, we definitely agree that there is something about memory going on here. We believe that part of the difference between 4 and 6-year-olds could be the result of developmental differences in spatial memory skills. We report this interpretation on page 28. Previous studies show that remembering the objects’ position in internal reference frames (like the one we are using here) is challenging for young children, so this factor may be telling in explaining the developmental differences we found. But, disentangling the specific role of memory skills from developmental differences in scaling abilities would require at least a comparison between map-reading tasks with and without the scaling component. We agree it would be important to run a study controlling for the memory factor, but the same it’s true for all the previous scaling studies with children. We hope that expanding the discussion of this central issue in the manuscript helps to clarify the results.

Some decimal points seem to be represented by a full stop, and some by a comma - these need checking for consistency

Answer: We tried hard to find those cases where the decimal point was represented by a comma, and we only find one case; we corrected it (Line 247). We would appreciate if you can highlight other cases for us to correct them.

6. PLOS authors have the option to publish the peer review history of their article (what does this mean?). If published, this will include your full peer review and any attached files.

Do you want your identity to be public for this peer review? For information about this choice, including consent withdrawal, please see our Privacy Policy.

Reviewer #1: No

Reviewer #2: No

---

## [Decision Letter · Decision Letter 1]

30 Nov 2020

Developmental differences in children’s and adults’ use of geometric information in map-reading tasks

PONE-D-20-10910R1

Dear Dr. Otálora,

I am pleased to inform you that after evaluation by two experts in the field and myself, your revised manuscript has been judged scientifically suitable for publication and will be formally accepted for publication once it meets all outstanding technical requirements.

Kind regards,

Sasha

Alexander N. 'Sasha' Sokolov, Ph.D.

Academic Editor

PLOS ONE

Additional Editor Comments (optional):

Reviewers' comments:

Reviewer's Responses to Questions

**Comments to the Author**

1. If the authors have adequately addressed your comments raised in a previous round of review and you feel that this manuscript is now acceptable for publication, you may indicate that here to bypass the “Comments to the Author” section, enter your conflict of interest statement in the “Confidential to Editor” section, and submit your "Accept" recommendation.

Reviewer #1: All comments have been addressed

Reviewer #2: All comments have been addressed

2. Is the manuscript technically sound, and do the data support the conclusions?

Reviewer #1: Yes

Reviewer #2: Yes

3. Has the statistical analysis been performed appropriately and rigorously? 

Reviewer #1: Yes

Reviewer #2: Yes

4. Have the authors made all data underlying the findings in their manuscript fully available?

Reviewer #1: Yes

Reviewer #2: Yes

5. Is the manuscript presented in an intelligible fashion and written in standard English?

Reviewer #1: Yes

Reviewer #2: Yes

6. Review Comments to the Author

Reviewer #1: "The authors thoroughly addressed the comments raised in both reviews. The current version provides a clear explanation of the rationale and a reasonable interpretation of findings; the design and analyses are particularly strong features of the study. My recommendation is to accept the manuscript."

Reviewer #2: This is a thorough revision of the original submission. The changes address all the comments that I made in my original review.

(n.b. there are some missing words on line178 which needs a minor correction).

7. PLOS authors have the option to publish the peer review history of their article (what does this mean?). If published, this will include your full peer review and any attached files.

Reviewer #1: No

Reviewer #2: No

---

## [Editor Report · Acceptance letter]

16 Dec 2020

PONE-D-20-10910R1 

Developmental differences in children’s and adults’ use of geometric information in map-reading tasks 

Dear Dr. Otálora:

I'm pleased to inform you that your manuscript has been deemed suitable for publication in PLOS ONE. Congratulations! Your manuscript is now with our production department. 

Kind regards, 

on behalf of

Dr. Alexander N. Sokolov 

Academic Editor

PLOS ONE